# The Influence of Tool Rake Surface Geometry on the Hard Turning Process of AISI52100 Hardened Steel

**DOI:** 10.3390/ma12193096

**Published:** 2019-09-23

**Authors:** Hanzhong Xu, Honggen Zhou, Zhengyu Ma, Lei Dai, Xuwen Jing, Guochao Li, Yujing Sun

**Affiliations:** 1School of Mechanical Engineering, Jiangsu University of Science and Technology, Zhenjiang 212000, China; xu_hanzhong@126.com (H.X.); 13505231579@163.com (Z.M.); 17802595281@163.com (L.D.); jingxuwen@just.edu.cn (X.J.); 2Jiangsu Provincial Key Laboratory of Advanced Manufacturing for Marine Mechanical Equipment, Jiangsu University of Science and Technology, Zhenjiang 212000, China; 3Joint Technology Transfer Center of Yancheng Polytechnic College, Yancheng Polytechnic College, Yancheng 224000, China; 4School of mechanical and automotive engineering, Qilu University of Technology (Shandong Academy of Sciences), Jinan 250000, China; syj2087@163.com

**Keywords:** hard turning, coordinating chip removal, finite element analysis, AISI52100 hardened steel

## Abstract

The hard turning process has been widely used in the field of hard material precision machining because of its high efficiency, low processing residual stress, and low environmental pollution. Due to its undesirably processing quality, it is still not a substitute for traditional grinding, so many studies have reported that the process has been optimized. However, there has been little research on the geometry optimization of hard cutting tools, which have a great influence on the traditional machining process. In this paper, two tools with different rake face shapes are designed. The finite element analysis method is used to compare their performance with a conventional plane tool while turning hardened steel. The results show that the cutting performance of the designed tool T1 and T2 (chip morphology, cutting force, and cutting temperature) and the quality of the machined surface are improved compared with the tool. The cutting force decreased by 12.72% and 14.74%, the cutting temperature decreased by 7.56% and 9.01%, respectively, and the surface residual stress decreased by 26.56% and 28.66%.

## 1. Introduction

Hardened steel is a typical wear-resistant and hard-to-cut material. Due to its high hardness and mechanical strength, it is widely used in industrial fields, such as in bearings and automobiles [1]. In general, the finishing of hardened steel processed by grinding. With the development of ceramic and polycrystalline cubic boron nitride (PCBN) cutting tools, hard turning has been widely used. Hard turning has many advantages over grinding, such as high productivity, high processing flexibility, high material removal rate, no use of coolant, and fewer environmental problems [2]. The main problem of hard turning is that the cutting performance of the tool and the surface quality of machined workpiece are poor, and the tool structure is one of the main factors affecting these problems. Cutting tool performance can be improved by changing the geometry of the cutting tool’s rake face to reduce the cutting force, cutting temperature, contact friction, and tool wear [3,4,5,6].

It has been found that changing the geometry of a tool could improve its cutting performance, including the geometry of the cutting edge, the geometry of the rake surface, and the micro-texture. Therefore, the structural design of tools has studied in machining. SHI et al. [7] changed the surface shape of the rake face by establishing a non-free cutting model, so the cutting deformation and chip flow of each part of the cutting edge are independent of each other when cutting, thereby increasing the chip removal capacity, allowing the chips to easily flow out from the rake face, and reducing the cutting force. Thiele et al. [8] studied the influence of tool cutting-edge geometry or edge preparation on residual stress in the hard turning of AISI52100 steel and concluded that in the precision hard turning of AISI52100 steel, the large edge hone tools produce deeper, more compressive residual stresses than small edge hone or chamfered tools. Yen et al. [9], by changing the shape of the cutting edge, studied the effects of the edge preparation of the cutting tool (round edge and chamfer edge) upon chip formation, cutting forces, and process variables (temperature, stress, and strain) in orthogonal cutting. Elbah et al. [10] studied hard turning AISI4140 steel with a wiper tool and a conventional tool and concluded that the wiper tool has better cutting performance, especially in terms of the surface roughness of the workpiece. Özel, T. [11] used 3D FE modelling to study the influence of variable edge micro-geometry of the polycrystalline cubic boron nitride (PCBN) tool on the cutting temperature during the turning of AISI4340. There are also many scholars to improve the cutting performance of the cutting tool by micro-texturing the tool’s rake surface. Zhang et al. [12] machined a micro-groove tool parallel to the main cutting edge on the rake face. When hard-hardening steel, the cutting force, friction coefficient, and surface roughness were significantly reduced compared with conventional tools. Xing et al. [13] studied the micro-scale texture of different geometric features prepared on the surface of the Al2O3/Tic ceramic tool and compared it with the cutting performance of a conventional tool in a dry cutting experiment, measuring tool wear, cutting force, cutting temperature, friction coefficient, surface roughness, and chip morphology. The results show that, compared with traditional tools, the micro-scale deformed tool significantly reduced the cutting force, cutting temperature, friction coefficient, and tool wear, and improved the surface roughness and quality of the machined workpiece. The reduction of cutting force and cutting temperature contributes to the service life of the tool and the strength of the cutting edge, which is of great significance in precision hard turning [14].

A large number of studies have pointed out that cutting parameters have a great influence on cutting performance. [15,16,17]. Shihab et al. [18] used a multi-layer coated carbide insert in hard-turn AISI52100 hardened steel to study the effects of different cutting parameters (*v_c_*, *f*, *a_p_*) on cutting performance. Bouchelaghem et al. [19] investigated the effects of cutting parameters on the hard turning of AISID3 (60 HRC) using the CBN tool. Their results showed that an increase in the value of the cutting parameters results in an increase in cutting temperature. Ebrahimi et al. [20] used statistical analysis methods to analyze the effects of different cutting parameters on cutting force, tool wear, and chip morphology. Bartarya et al. [21] used the uncoated CBN tool with a honed edge to hard-turn the AISI52100 hardened steel and found that the cutting parameters (*a_p_*, *f*) have a great influence on the cutting force. Alok et al. [1] used a coated carbide tool to hard-turn AISI52100 steel, establishing the input process parameters (*v_c_*, *f*, *a_p_*) and output response (main cutting force, radial force and feed force, maximum side grinding volume, and the surface quality of the workpiece), the influence of the cutting parameters on cutting performance was studied by experimental statistical design.

FE method simulation has been widely used to predict the performance of machining processes and has shown good results [22,23,24,25]. Kim et al. [26] used Deform 3D simulation to analyze the influence of tool rake surface texture design on the cutting force, effective friction coefficient, and chip flow direction during the hard turning of hardened steel (AISI52100). The vertical texture, parallel texture, and rectangular texture designed on the rake face, and the cutting performance, were analyzed compared to the unstructured tool. The results show that the cutting edge and the friction coefficient of the rake face micro-texture tool are smaller than those of the unstructured tool, and the flow of the chip is small. Research has also shown that the flow direction of the chip is related to the shape and size of the texture on the rake face of the tool. Coroni et al. [27] used Deform 2D to predict the optimal experimental conditions for orthogonal machining of various titanium alloys, and the simulation results agree well with the experimental results. Mishra et al. [28] used AdvantEdge FE simulation software to study the effects of different texture shapes (circles, squares, triangles, and ellipses) on the cutting force of titanium alloys during processing under dry conditions, as well as different texture areas, densities, and the effect of depth on changes in the cutting force. It has been found that the effect of texture shape is found to be less influential for dry cutting. However, area density is found to have the most dominant effect on cutting forces. Liu et al. [29] used the AdvantEdge FE simulation software to study the cutting performance of the curved microgroove tool, and it was found that the newly designed microgroove tool is superior to the non-deformed microgroove linear microgroove tool in terms of tool-chip interface friction, chip formation, cutting force, cutting temperature, and tool stress.

It can be seen from previous studies that changing the shape and texture design of the rake face can improve the tribological performance of the tool-chip interface by reducing the frictional force, the cutting force, and the cutting temperature. However, few studies have been conducted on reducing the friction by affecting the chip’s removal. In this paper, different surface shapes were designed on the rake face to coordinate the chip removal, so the chips can easily flow out from the rake face, thereby reducing the tool–chip interface’s friction. We also carried out a 3-D Finite element (FM) simulation study on its cutting performance. The effects of different rake face tools and traditional planar tools and cutting parameters on the chip morphology, cutting force, cutting temperature, and residual stress distribution on the workpiece surfaces were compared. In order to improve the cutting performance of the tool and hard cutting surface quality lay a foundation.

## 2. Materials and Methodology

### 2.1. Experimental Research

In order to study the influence of the geometry of the tool rake face on the turning process, this paper reduces chipping interference by designing different rake face tools to improve the chip’s discharge capacity, thus reducing the friction between the tool and chip interface to improve the cutting performance of the tool. Surface tools were designed using the UG12.0 software (Siemens PLM Software, Berlin, Germany) and imported into the FE model at the same tool holder angle. The geometric parameters of the tool are shown in Table 1. The traditional plane tool is TN, and the two newly designed tools are T1 and T2. The main difference between the newly designed tool T1 and T2 is that the concave and convex degrees of their grooves are different. The different rake face tool models and their specific surface parameters are shown in Figure 1.

In order to compare the influence of different rake face geometries and traditional planar tools on the cutting performance of AISI52100 hardened steel, a turning test of different cutting parameters is studied in this paper. The cutting parameter test shown in Table 2 was performed using the conventional plane tool TN and the newly designed tools, T1 and T2 [30,31,32]. This test is designed for an orthogonal test design, and the L9 (3^4^) orthogonal array is selected for the experimental design. The influences of the cutting speed *v_c_*, feed *f*, and cutting depth *a_p_* on the cutting performance of the different cutters are analyzed, and the number of test designs was reduced.

### 2.2. Finite Element Modeling and Verification

As can be seen from the typical turning geometry model in Figure 2a, the cutting force of the turning process is generated by three directions (*X*, *Y*, *Z*), which are the main cutting force-*F_x_*, the radial force-*F_y_*, and the axial force-*F_z_*. This paper uses the AdvantEdge software (Third Wave Systems, Minneapolis, MN, USA) for turning simulation. A Model of AISI52100 hardened steel workpiece was established with a diameter D of 6 mm and a length L of 3 mm. Its chemical composition is shown in Table 3, and its physical properties are shown in Table 4. This model was then combined with the cutting tool to form the turning simulation model, as shown in Figure 2b. Among the directions, the *X* direction is the cutting direction, the *Y* direction is the cutting depth, and the *Z* direction is the direction of feeding.

The constitutive model used in this paper is the Johnson–Cook (J-C) model proposed by Johnson and Cook [33,34], which describes the flow stress of the product material of the strain, strain rate, and temperature effects. In recent decades, the Johnson–Cook (J-C) model, as shown in Equation (1), has been widely used in FE machining simulations [35], and its constitutive parameters are shown in Table 5 [36]:(1)$$\bar{\sigma }=(A+B{\bar{\varepsilon }}^{n})[1+C\ln(\frac{\overset{\cdot }{\bar{\varepsilon }}}{\overset{\cdot }{\bar{\varepsilon_{0}}}})][1-{\left(\frac{T-T_{o}}{T_{m}-T_{o}}\right)}^{m}]$$
where $\bar{\sigma }$ is the flow stress, $\bar{\varepsilon }$ is the plastic strain, $\overset{\cdot }{\bar{\varepsilon }}$ is the plastic strain rate, and $\overset{\cdot }{\bar{\varepsilon_{0}}}$ is the reference plastic strain rate. *T*, *T_0_*, and *Tm* are the work temperature, reference temperature, and material melting temperature, respectively. *A*, *B*, *n*, *C*, and *m* are the material constants.

This FE model adopts adaptive mesh generation technology, and the mesh density of the cutting area is relatively large near the plastic deformation area, while the mesh density of the other areas is relatively small, so a better cutting area grid is obtained. The maximum unit size of the workpiece is 1 mm, and the minimum unit size is 0.15 mm. The minimum unit side length of the cutting edge is 0.01 mm, and the radius of refined region of the cutting edge is 0.05 mm. In order to better observe the variation and contrast of the chip’s morphology, the minimum unit’s side length for the chip is 0.01 mm, and the chip’s refinement factor is 3.

The FE model for the roller turning was established and verified for the processing technology of the roller. The metal cutting process was complex, and there were many factors affecting the turning model. In order to obtain an accurate 3-D turning model, the model was verified by chip morphology and surface residual stress [37].

In the process of metal cutting, the shape of metal chips is often a direct reflection of the cutting effect, and the characteristics of these chips play a crucial role in surface integrity and tool state [38]. The FE simulation of the turning roller was carried out by the AdvantEdge software. The diesel engine roller was made of AISI52100 hardened steel, with a Rockwell hardness of 60 HRC. The CNC machining center used in our machining was CK0625, and the workpiece was AISI52100 hardened steel (HRC 60). The tool used in the actual test is shown in Figure 3a, and the material of the tool was Al_2_O_3_–Tic. The composition and mechanical properties are shown in Table 6. Both the actual machining and the simulation machining used the same cutting parameters: cutting speed *v_c_* = 150 m/min, feed *f* = 0.15 mm/r, and cutting depth *a_p_* = 0.36 mm. The chip obtained by simulation is shown in Figure 3b. As shown in Figure 3c, the chip is a spiral, unlike the experimentally obtained chip, which is basically the same as the simulation chip, thereby verifying the reliability of the simulation model.

The equipment used to test the residual stress was an X stress 3000 G2, an X-ray stress tester from STRESSTECH OY, Jyväskylä, Finland. The surface of the hardened steel workpiece of the AISI52100 was divided into four points along the circumference. A point selection test was carried out, as shown in Figure 4a, and the mean value was taken after measurement. The test equipment and test procedure are shown in Figure 4. A comparison of the residual stress obtained by the simulation and the experiment is shown in Figure 5. Under different cutting conditions, the difference of residual stress is less than 15% and has good consistency.

In summary, the cutting effect of the three-dimensional FE model established in this paper is in good agreement with its actual cutting effect, which can be used to study the influence of different cutting surfaces on cutting performance.

## 3. Results and Discussion

The simulation model uses the validated 3D model established in Section 2.2. The material of the tool is the same as that of the actual ceramic tool, as shown in Table 6. The material of the workpiece is the same as the material properties of the actual ASISI52100, as shown in Table 3 and Table 4. The test was designed with an orthogonal test design, and the L9 (3^4^) orthogonal array was chosen for the experiment. The cutting conditions of the three tools are the same, using the orthogonal test tables shown in Table 2.

### 3.1. Influence of the Curved Tool on Chip Morphology

Chip generation has a significant impact on a workpiece surface’s finish, tool life, and overall cutting operations [39]. Liu et al. [25] studied the 3D chip’s morphology during the 17-4PH orthogonal cutting process. The influence of different microgroove tools on chip curling is illustrated by analyzing the degree of chip curling after cutting. Defining a method for measuring the degree of chip curl was used to measure the radius of the bend based on chip formation. A larger radius indicates a smaller degree of curling, and the opposite indicates a greater degree of curling.

Figure 6 shows the chip morphology generated by the different cutting tools designed in this paper under different cutting parameters. According to Figure 6, compared with the traditional plane tool, the chip processed with the newly designed tools (T1 and T2) has slight bending. Radius r, which measures the chip curl, is used to indicate the degree of curling of the chips under different cutting conditions. This method is shown in Figure 7. During the process of turning machining, the contact friction between the tool and the chip causes the chip to curl and form an arc, and the inner circle measures the radius *r*. The measured average-bending radius of each group of test chips is shown in Figure 8, figure Δ*A_r_*, and Δ*B_r_*, respectively for the T1 and T2 tools. For the relative difference values of the tools relative to the traditional TN, the calculation formula is shown in Equation (2):(2)$$\Delta A_{r}=\frac{TN-T1}{TN},\;\Delta B_{r}=\frac{TN-T2}{TN}.$$

According to the comparison of the chip bending radius generated by different tools under different cutting parameters, the curled radius was reduced by a maximum of 16% and 20.8% when cutting with the newly designed T1 and T2 tools compared to the conventional planar tools, while the average curl radius was reduced by 10.95% and 15.54%. The smaller the curl radius of the chip, the more severe the chip curl, which proves that the curved surface design of the rake face promotes the curling of the chip. This is due to the reduced contact area of the tool–chip interface, which reduces the frictional force between the chips and causes the chips to curl easily.

The single factor influence analysis of the cutting parameters in Figure 9 shows that the cutting depth *a_p_* and the feed amount *f* have the greatest influence on the chip curl’s radius, and the cutting speed *v_c_* has the least influence. The curl radius of the chip increases with the depth of the cut and the amount of feed. This is because when ap and f increase, the contact area between the tool and the chip increases, and the friction between the tool and the chip increases. This way, the chips are not easy to curl.

The newly designed curved tool can make the chip discharge better than the traditional plane tool, reduce the frictional force between the tool and the chip, reduce the chip curl’s radius, and generate more chips. This design can also more efficiently remove the cutting heat from the workpiece’s surface, which is very beneficial to the surface quality of the workpiece.

### 3.2. Influence of the Curved Tool on the Cutting Force

As shown in Figure 10, we conducted a comparison of the cutting force between the traditional plane tool and the newly designed curved tools under different cutting parameters. Under all cutting parameters, the cutting force of the newly designed curved tools was significantly lower than that of the conventional plane tool. Figure Δ*A_F_* and Δ*B_F_*, respectively, for the T1 and T2 tools, show the difference value of the tools relative to the traditional TN. The calculation formula is shown in Equation (3):(3)$$\Delta A_{F}=\frac{TN-T1}{TN},\;\Delta B_{F}=\frac{TN-T2}{TN}.$$

By comparison, the cutting force of the T1 tool is reduced by a maximum of 28.91% compared with the conventional plane tool, while the cutting force of the T2 tool is reduced by a maximum of 31.62% compared with the conventional plane tool. The newly designed T1 and T2 cutters produce an average of 12.72% and 14.74% less cutting force than conventional planar tools, which improves the wear resistance and service life of the tools. When using the newly designed curved surface tool, the contact area of the tool–chip interface is significantly reduced. Moreover, the design of the rake face reduces the chipping interference to a certain extent, thereby reducing the friction between the chip and the rake face of the tool. This is the main reason for the reduction of the cutting force. From the curl radius of the chip, it can be seen that the curved tool is prone to chipping, so that the contact length between the tool and the chip also decreases, which leads to a reduction in the cutting force.

According to the single factor influence of the cutting parameters in Figure 11, the cutting speed has little influence on the cutting force, while the cutting depth *a_p_* and the feed *f* have great influence on the cutting force, which increases with an increase in cutting depth and feed. This result is due to the increased contact area of the tool–chip interface as the feed rate and depth of cut increase. This increases the friction between the chip and the rake face of the tool, resulting in an increase in cutting force.

In general, the two newly designed curved tools reduce the cutting force compared to conventional planar tools, which lays the foundation for the reduction of the surface residual stress caused by mechanical stress.

### 3.3. Influence of the Curved Tool on the Cutting Temperature

For studying the influence of the surface of the newly designed curved tool on the temperature of the processed surface, the cutting temperature of 60 points on the processed surface were extracted. This extraction method is shown in Figure 12, and the mean value was taken as the cutting temperature of the processed surface. Three measurements were taken for each machined surface, and the average temperature was calculated. The influence of the different curved tools on the cutting temperature is shown in Figure 13. It can see that under all cutting parameters, the cutting temperature of the newly designed curved tool is smaller than that of the conventional plane tool. Figures Δ*A_T_* and Δ*B_T_*, respectively, for the T1 and T2 tools show the difference value of the tool relative to the traditional TN. The calculation formula is shown in Equation (4):(4)$$\Delta A_{T}=\frac{TN-T1}{TN},\;\Delta B_{T}=\frac{TN-T2}{TN}$$

In the test, the newly designed T1 and T2 tools maximally reduced the surface temperature generated by cutting better than the traditional planar tools. The max reduction was 11.31% and 14.02%, while the average was 7.56% and 9.01%. Contact friction between the tool and the chip is one of the main sources of chip heating. Compared with the conventional plane tool, curved tools have a small contact area between the chip and the rake face of the tool after cutting, which reduces friction. This is the main reason for the reduction in cutting temperature. Due to the obvious chip curl caused by the curved surface of the rake face, the contact length and friction between the tool and the chip also changed, thereby reducing the cutting temperature of the processed surface. The contact area between the tools and chips was reduced, thus causing a reduction of the heat transfer between the workpiece and the tool. The heat generated in the primary deformation zone was mainly removed by the chips instead of being transmitted to the cutting tools and, further, to the machined surfaces. As can be seen in Section 3.1, the curved tools can produce more chips and take more cutting heat. The grooves on both sides of the curved tools also provide greater passage for the cutting heat. As shown in Figure 10, the reduction in cutting force indicates that the energy consumed by the newly designed curved tools’ cutting process was also reduced, which also reduces the generation of cutting heat.

The single factor effect of the cutting parameters on cutting temperature can see in Figure 14. The depth of the cut and the amount of feed *f* have a great influence on the cutting temperature, which is consistent with the conclusions drawn by Bouacha et al. [15]. Moreover, the cutting temperatures of the three tools increase with an increase in the feed rate and depth of the cutting. This shows that when the cutting amount is small, the influence on the cutting temperature is also small, and when the cutting amount increases, the influence also increases. The curved surface design of the rake face also has a great influence on the cutting temperature.

In general, the two curved tools of the new design have a much lower cutting temperature than conventional planar tools, providing support for reducing the surface residual stress caused by thermal stress.

### 3.4. Influence of the Curved Tool on Residual Stress

Residual stress on the surfaces of the parts after cutting is mainly caused by plastic deformation (extrusion and tensile), thermoplastic deformation, and phase change stress [40]. The influence of the newly designed T1 and T2 tools on the surface residual stress of the AISI52100 hardened steel workpiece under different cutting parameters was studied and compared with the traditional plane tool. The residual stress value of the machined surface was obtained through simulation. As shown in Figure 15, nine test points were obtained by dividing three lines horizontally and longitudinally on the machined workpiece surface, the residual stresses of the these points were selected, and the averages were taken as the residual stresses of the machined surface. The test results are shown in Figure 16. Figure Δ*A_σ_* and Δ*B_σ_* respectively for the T1 and T2 tools show the difference value of the tool relative to the traditional TN. The calculation formula is shown in Equation (5):(5)$$\Delta A_{\sigma }=\frac{TN-T1}{TN},\;\Delta B_{\sigma }=\frac{TN-T2}{TN}.$$

According to the comparison of the residual stress on the cutting surface of the T1/T2 and TN tools, it can see that the residual stress generated by the cutting surface of the T1 tool and T2 tool is reduced by a maximum of 36.98% and 48.24%, and 26.56% and 28.66%, on average. This is due to the reduced friction of the newly designed curved tool and the cutting surface, which reduces the plastic deformation caused by mechanical stress. Along with the reduction of the surface temperature studied in Section 3.3, the plastic deformation caused by thermal stress is reduced, and, finally, the residual stress of the surface is reduced [41].

This paper not only considers the influence of a single factor on residual stress but also considers the influence of cutting parameter interactions on residual stress. Figure 17 shows the influence of the single factor cutting speed *v_c_*, feed *f*, and depth of cutting *a_p_* on residual stress. The cutting speed has a great influence on the surface residual stress, and its value decreases with an increase of its cutting speed. However, *f* and *a_p_* have little influence on residual stress, which decreases slightly with an increase in feed and cutting depth. According to the interaction diagram in Figure 18, the residual stress first rises and then declines with an increase in feed and cutting depth, forming a peak when the feed is 0.16 mm/r and the cutting depth is 0.24 mm. The residual stress on the surface of the AISI52100 hardened steel is turned by different cutting parameters with different cutters, and the residual compressive stress is generated on the surface of the workpiece. With an increase of the cutting speed, the residual stress on the surface decreases. This is because the surface temperature of the workpiece decreases while the cutting speed increases, thereby reducing the residual stress caused by thermal stress. With the increase of feed, the residual stress of the machined surface decreases, which is caused by the reduction of mechanical stress and plastic deformation. The residual compressive stress produced by the surface can improve the fatigue life of the material. Therefore, in order to improve the surface quality of the AISI52100 hardened steel in hard turning, it is also necessary to consider the cutting parameters.

## 4. Conclusions

The designed tool with a curled rake face exhibited good cutting performance in the hard turning of the AISI52100 hardened steel. Compared with the traditional flat tool TN, the newly designed tools, T1 and T2, have a cutting force reduction of 12.72% and 14.74 and a cutting temperature of 7.56% and 9.01%. The surface residual stress is reduced by 26.56% and 28.66%.In addition, the influence of cutting parameters on the cutting performance of the tool is also studied. The feed rate and depth of cut have a great influence on the chip’s shape, cutting force, and cutting temperature. The cutting force and cutting temperature increase with an increase of the feed rate and the depth of cut. The cutting speed has a great influence on the surface residual stress, and the surface residual stress decreases with an increase of the cutting speed.The residual stress first rises and then declines with an increase of the feed and cutting depth, forming a peak when the feed is 0.16 mm/r, and the cutting depth is 0.24 mm.During our research, the manufacturing methods for the design tools were not provided but will be discussed in further studies.

## Figures and Tables

**Figure 1 materials-12-03096-f001:**
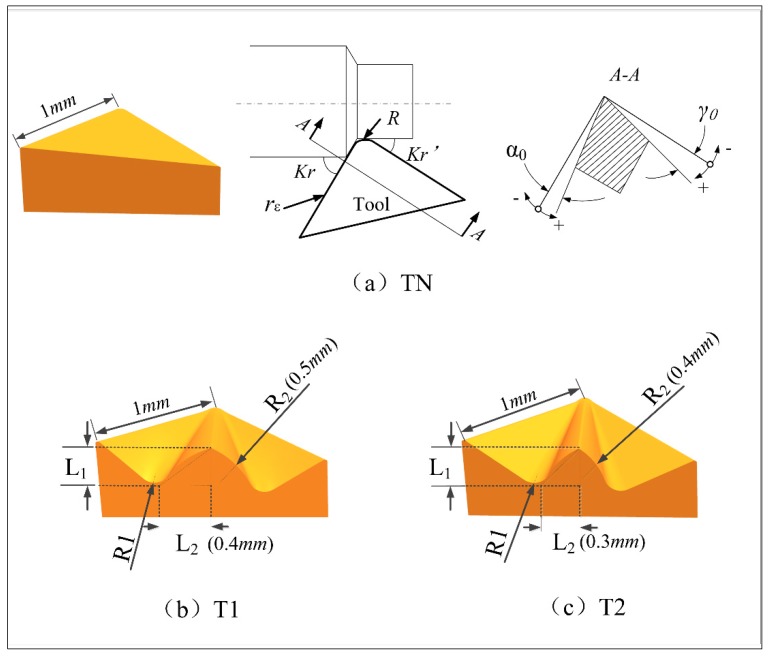
Different surface tool models and their surface parameters: (**a**) Plane tool-TN, (**b**) curved tool-T1, and (**c**) curved tool-T2.

**Figure 2 materials-12-03096-f002:**
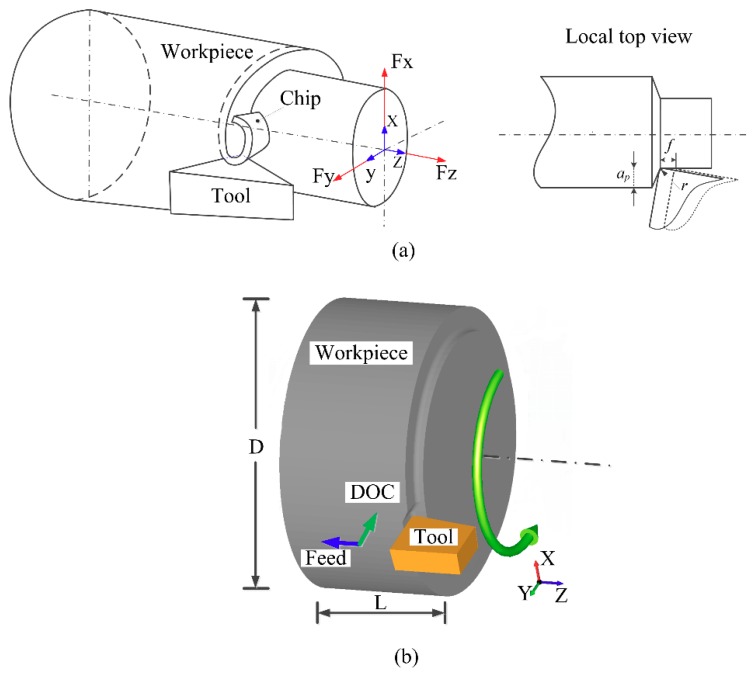
(**a**) Turning geometry model; (**b**) 3-D turning finite element model.

**Figure 3 materials-12-03096-f003:**
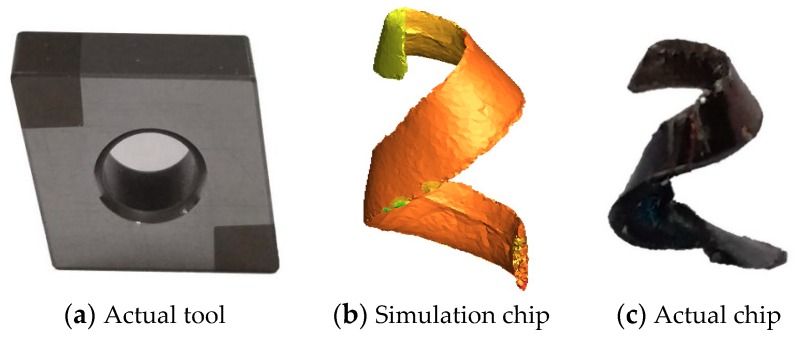
Simulation chip and actual chip comparison.

**Figure 4 materials-12-03096-f004:**
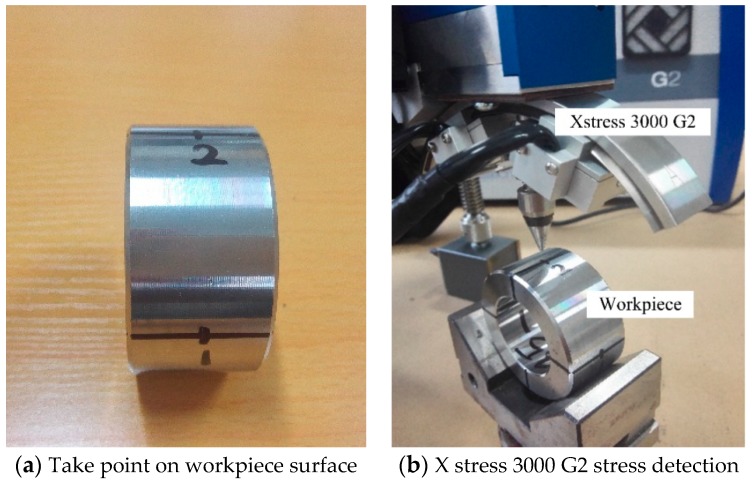
Residual stress testing equipment and testing process.

**Figure 5 materials-12-03096-f005:**
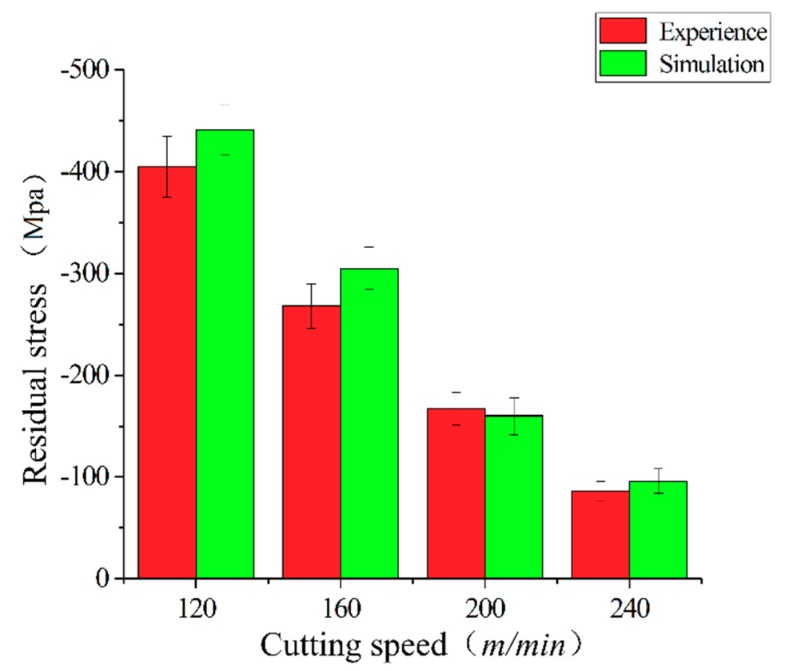
Test measurement of the residual stress and simulation results.

**Figure 6 materials-12-03096-f006:**
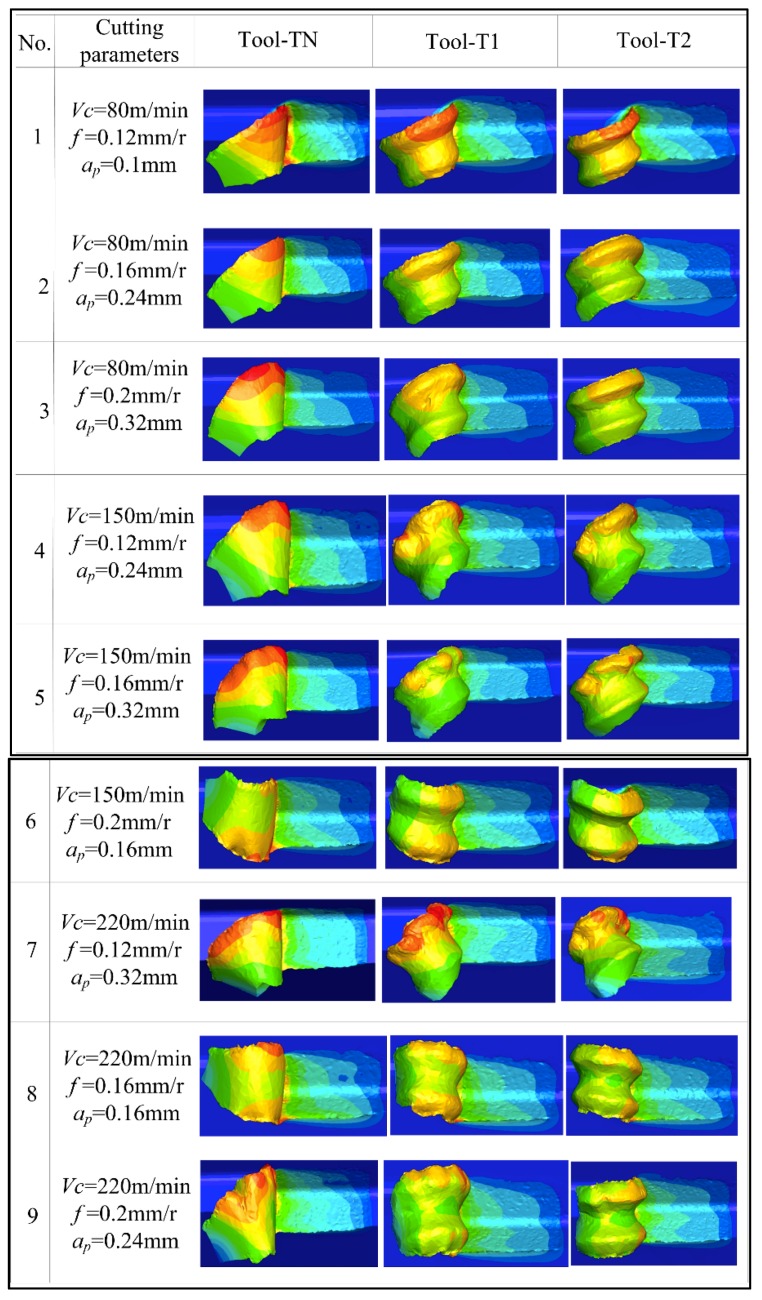
Chip morphology of different curved tools under different cutting parameters.

**Figure 7 materials-12-03096-f007:**
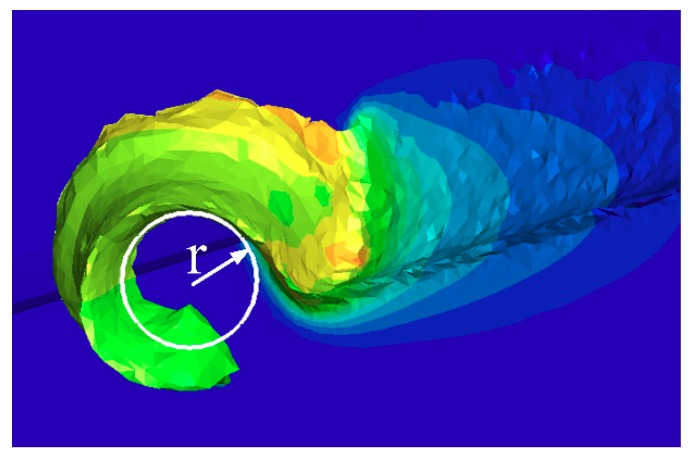
Measurement of the chip’s bending radius.

**Figure 8 materials-12-03096-f008:**
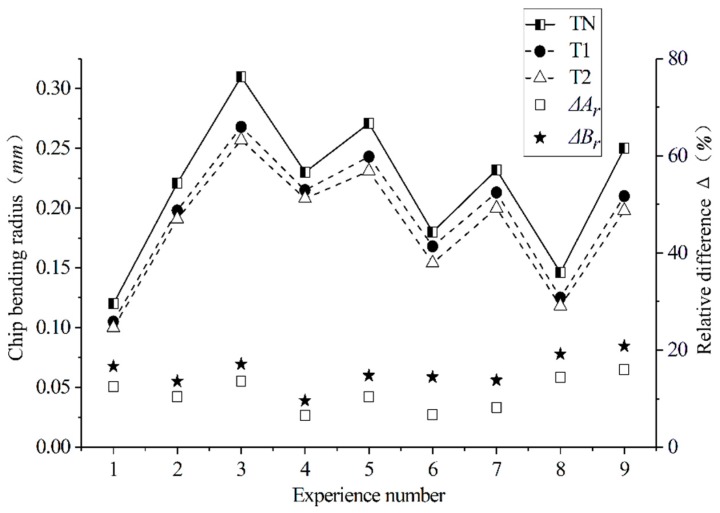
Comparison of the chip curl radius generated by the different cutting parameters of different tools.

**Figure 9 materials-12-03096-f009:**
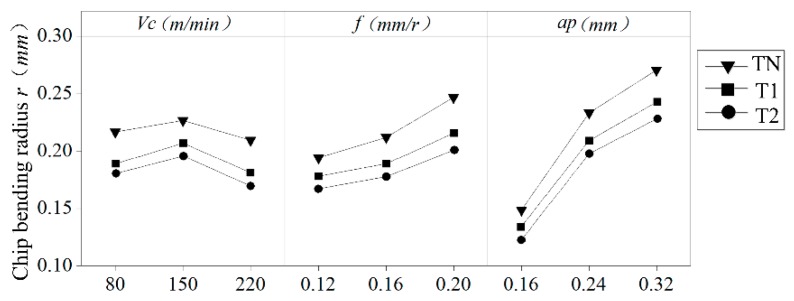
The single factor effect of the cutting parameters on the chip curl radius.

**Figure 10 materials-12-03096-f010:**
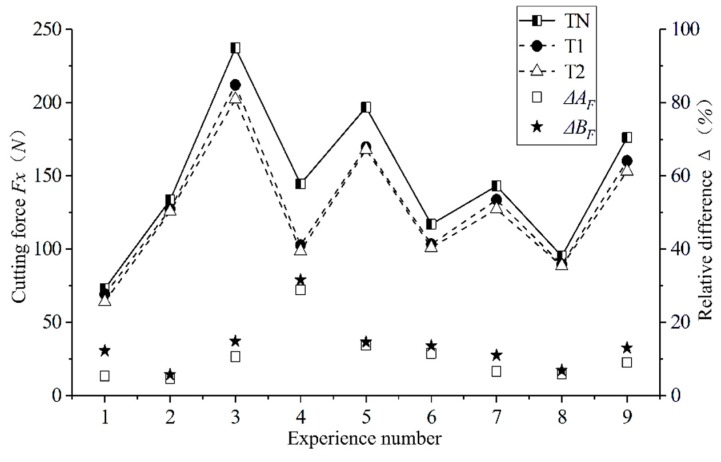
Cutting force comparison of different cutting parameters with different cutting tools.

**Figure 11 materials-12-03096-f011:**
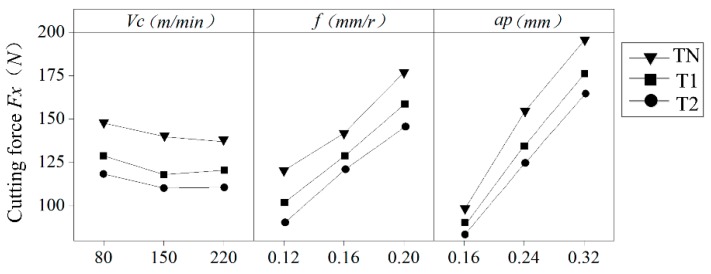
Single factor effect of cutting parameters on cutting force.

**Figure 12 materials-12-03096-f012:**
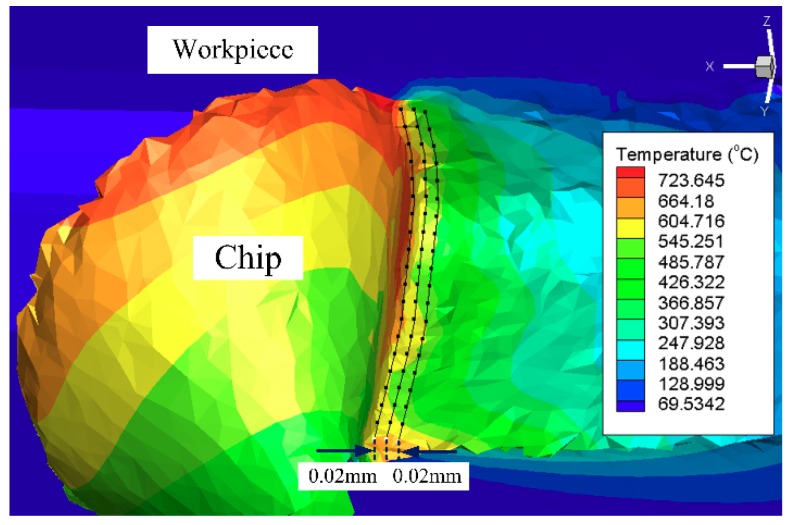
Measuring cutting temperature.

**Figure 13 materials-12-03096-f013:**
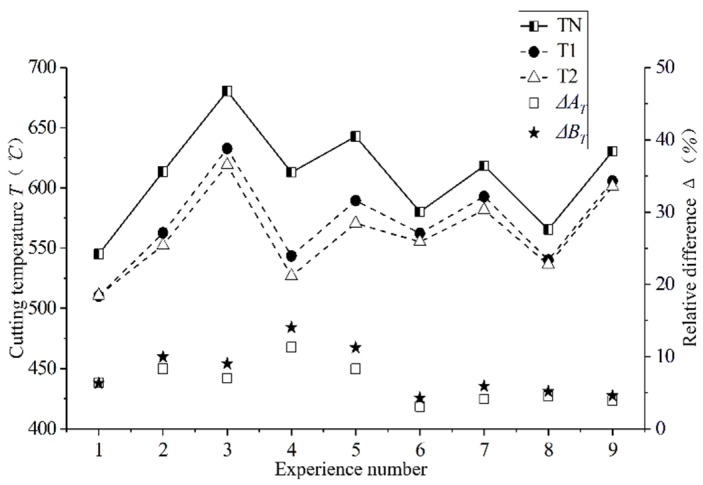
Cutting temperature comparison of different cutting parameters with different cutting tools.

**Figure 14 materials-12-03096-f014:**
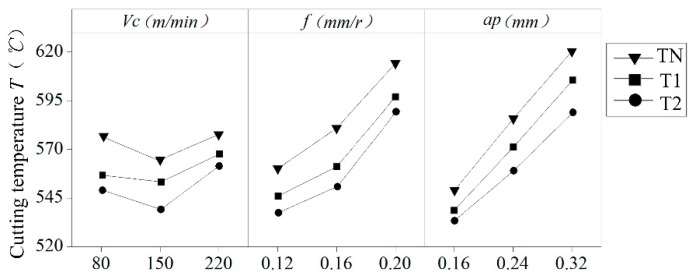
Single factor effect of cutting parameters on cutting temperature.

**Figure 15 materials-12-03096-f015:**
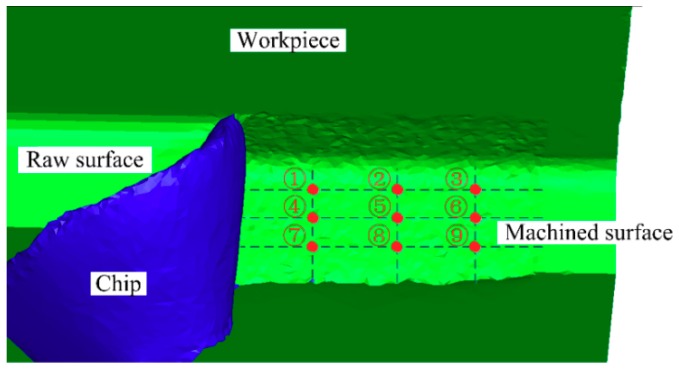
Surface residual stress test point selection (1–9) measurement.

**Figure 16 materials-12-03096-f016:**
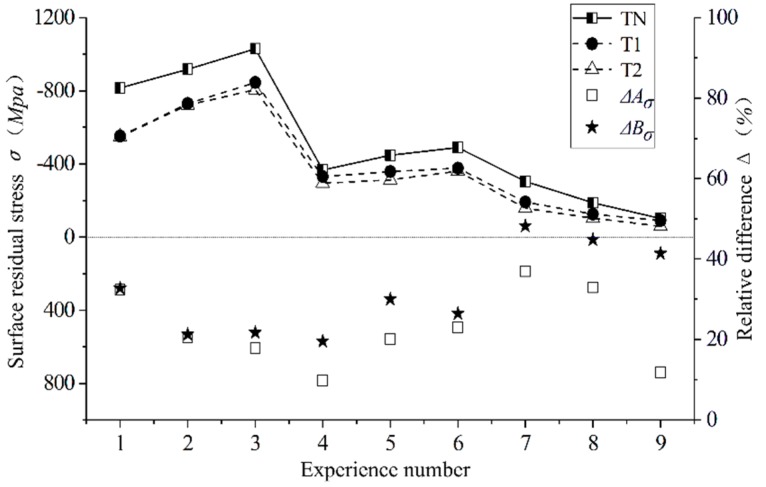
Surface residual stress under different cutting parameters with different tools.

**Figure 17 materials-12-03096-f017:**
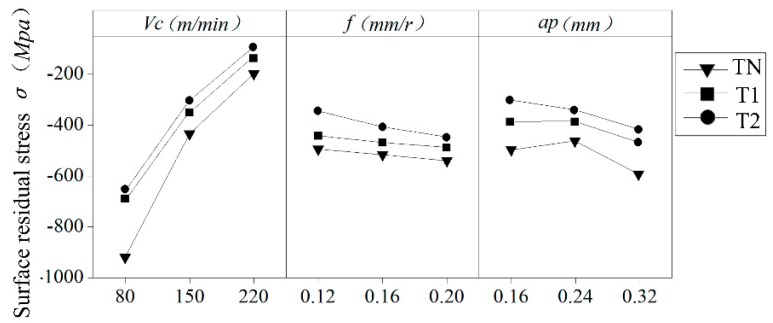
Single factor effect of cutting parameters on the surface residual stress.

**Figure 18 materials-12-03096-f018:**
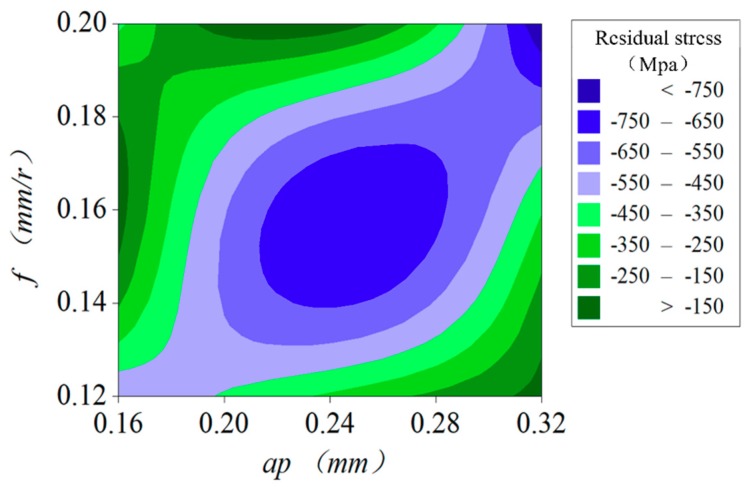
Interaction of the cutting parameters on surface residual stress.

**Table 1 materials-12-03096-t001:** Geometry of the tool.

Distinction	Geometric Parameters	Tool-N (TN)	Tool-1 (T1)	Tool-2 (T2)
Commons	rake angle $\gamma_{0}$	7°		
	relief angle $\alpha_{0}$	3°		
	nose radius R	0.1 mm		
	cutting edge radius $r_{\varepsilon }$	0.01 mm		
	inclination angle λs	0°		
	tool cutting edge angle Kr	90°		
	end cutting edge angle Kr*’*	0°		
	curved surface height L1	0.3 mm		
	bottom groove radius R1	0.1 mm		
Differences	curved surface width L2	-	0.4	25%*
	curved surface radius R2	-	0.5	20%*

Note: $25\%*=\frac{0.4-0.3}{0.4}$, $20\%*=\frac{0.5-0.4}{0.5}.$

**Table 2 materials-12-03096-t002:** Level design of the orthogonal test factors.

Level	*Vc* (m/min)	*f* (mm/r)	*a_p_* (mm)
1	80	0.12	0.16
2	150	0.16	0.24
3	220	0.2	0.32

**Table 3 materials-12-03096-t003:** Major chemical components of AISI52100 hardened steel.

Chemical Composition (%)					
C	Cr	Mn	P	S	Si
1.04%	1.45%	0.35%	0.025%	0.025%	0.23%

**Table 4 materials-12-03096-t004:** Physical properties of the AISI52100.

Density (kg/cm^3^)	Poisson’s Ratio	Young’s Modulus (GPa)	Thermal Conductivity (W/m/°C)	Specific Heat (J/kg/°C)	Expansion (μm/m/°C)
7.8	0.3	210	45	477	1.2

**Table 5 materials-12-03096-t005:** **The** Johnson–Cook (J-C) constitutive model coefficient for AISI52100.

*A*/MPa	*B*/MPa	*n*	*C*	*m*	*T_m_*
2482.4	1498.5	0.19	0.027	0.66	1487

**Table 6 materials-12-03096-t006:** Composition and mechanical properties of the ceramic tools.

Main Ingredient	Density (g/cm^3^)	Hardness (HRA)	Bending Strength (MPa)	Impact Toughness (J/cm)	Heat Resistance (°C)
Al_2_O_3_ + TiC	4.5	93.5~95	>800	4.7	1200~1300

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
