# Peer review of "The Influence of Tool Rake Surface Geometry on the Hard Turning Process of AISI52100 Hardened Steel"

_materials, 2019, doi:10.3390/ma12193096_

Round 1
Reviewer 1 Report
Paper title:
Influence of Tool Rake Surface Geometry on the Hard Turning Process of AISI52100 Hardened Steel
The coauthors of this paper presented a model to optimize the cutting tool geometry during hard turning. This work needs some improvements before being accepted.
The following points are to be considered:
Line 19, " well-machined residual stress ", this is not an adequate expression. Line 38, " The key challenges of hard turning are the performance of the tool and the poor quality of the processing surface, and the tool structure is one of the main 39 factor affecting the processing", please reconstruct the sentence. Line 48, "reducing the chip evacuation interference", not correct expression. Line 101, "by reducing friction", which friction: friction coefficient or frictional force? Line 102, "cutting heat", better to be "cutting temperature". Line 110, "reducing the friction", please refer to comment # 3 Why the shapes of T1 and T2 were selected? please mention the reasons. Tool material is not considered in section 2, it has to be mentioned and considered in the paper. Tool wear has to be considered in the experimental work, it should be limited to a certain value for not affecting the obtained results. Workpiece material hardness was not mentioned in section 2. It is not clear to the reviewer how the simulated results in figure 6 were obtained. Line 219, "reducing the friction" please refer to comment # 3. This has to be considered in the whole manuscript. Figure 7, can the coauthors add images for the actual chips? Line 242, reducing the cutting forces does not increase the strength of the cutting tool. Lines 251- 252, the mentioned reason for reducing the cutting forces is not accurate. Lines 270-271, frictional action is not the main reason for generating the heat during cutting, please correct. Lines 312-315, please refer to a good reference to justify your results. The conclusion section has to be rewritten depending on the above-mentioned results.
Author Response
Dear Editors and Reviewers:
Thank you for your letter and for the reviewers’ comments concerning our manuscript entitled “Influence of Tool Rake Surface Geometry on the Hard Turning Process of AISI52100 Hardened Steel” (ID: materials-580481). Those comments are all valuable and very helpful for revising and improving our paper, and of significance to our research. We have studied the comments carefully and have made corrections which we hope meet with approval. And we have also made a lot of changes to the English quality of this article. Revisions are marked in red in the paper. Please refer to the attachment for the main changes in this article and the responses to reviewers' comments.
We appreciate for Editors/Reviewers’ warm work earnestly, and hope that the correction will meet with approval.
Once again, thank you and all the reviewers for the kind advice.
Sincerely yours,
Hanzhong Xu

Reviewer 2 Report
The reviewer comments of the paper «Influence of Tool Rake Surface Geometry on the Hard Turning Process of AISI52100 Hardened Steel»
- Reviewer
The authors presented an article about an influence of tool rake surface geometry on the hard turning process of AISI52100 hardened steel. In general, the article is well written and deserves attention. However, there are several points in the article that require further explanation.
Comment 1:
Abstract as a whole sounds good. Add the name and model of the material for which the turning was carried out in the article. Demonstrate the novelty of the article and the advantages of the proposed method. What is the difference from previous work in this area? Show practical relevance.
Comment 2:
The introduction is written very well and clearly. However, at the end of the introduction it is necessary to clearly state the purpose of this article.
Show the relevance of the selected workpiece material from AISI52100. Where is it used? Give some cozier quotes on grinding this material.
It will also be useful to include a number of important articles on the topic in the introduction:
Characterization and modelling of the residual stresses induced by belt finishing on a AISI52100 hardened steel. J. Mater. Process. Technol. 2008, 208(1-3), 187-195. 10.1016/j.jmatprotec.2007.12.133
Effect of cutting parameters on cutting force and surface roughness during finish hard turning aisi52100 grade steel. Procedia CIRP 2012, 1(1), 651-656. doi:10.1016/j.procir.2012.04.116
Influence of different grades of CBN inserts on cutting force and surface roughness of AISI H13 die tool steel during hard turning operation. Materials 2019, 12(1), 177. doi:10.3390/ma12010177
Comment 3:
Section 2. Tool and test design and 3.1 3-D Finite element modeling combine into one with the title 2. Materials and methods.
Comment 4:
Describe the technology of obtaining different of Tool Rake Surface Geometry. What equipment was used? What accuracy and roughness are obtained of Tool Rake Surface Geometry?
Comment 5:
Give a tables with the chemical composition of the material of cutting tool.
Comment 6:
Explain and justify the geometric parameters and shapes presented in table 1.
Comment 7:
Give the text of the name of the articles of all physical quantities, for example in tables and figs. Why are these cutting conditions accepted for research?
Comment 8:
In section 2, specify the model of the CNC machine.
Comment 9:
How many replicates of the experiment were used in the studies? What is the reliability of the experimental results?
Comment 10:
Replace in figs. 9, 10, 11, 12, 14, 15, 17 and 18 are black and white on colored lines.
Comment 11:
The results obtained are shown in Fig. 6-18 should be explained in more detail in the text of the relevant sections. At least one two sentences for each drawing. Describe in the figure captions to Figs. 6-18 cutting conditions for which dependencies are built.
Comment 12:
Make a brief summary of the sections 4.1, 4.2, 4.3.
Comment 13:
It will be useful to add a section of Nomenclature in which to sign all the physical quantities and abbreviations encountered in the article. There are many physical quantities in the text and such a section will help to find the description of the necessary element.
For example,
ap : Cutting depth (mm)
etc.
Comment 14:
The conclusions in the article are generally good. What exactly was the design of the tool? But more clearly show the novelty of the article and the advantages of the proposed method. What is the difference from previous work in this area? It would be nice to compare the results of the article with turning with standard smooth cutting inserts. Show practical relevance. Conclusions should be relevant to the intended purpose of the article.
In general, the topic of the article are relevant. The article is interesting and useful, but needs to be improved. I recommend this article for publication in journal "Materials" after major changes.
Author Response

(The authors gave the same response as above.)

Reviewer 3 Report
1. Abstract.
Please do not include generally known process descriptions in the summary. The summary is to briefly present what is contained in the publication. Discussion of the main results.
2. Keywords.
We place the keywords in order from most important to less important. They should be more precise. They are too general.
3. Tool and test design.
- better to name this chapter: research methodology 3.1. Experimental research. 3.2. Numerical research. The current fragmentation of chapters is bad. Some information on the tested materials, tool shape, chemical composition of the material, etc. - should be together in chapter 3.1. Information on the FEM model itself, boundary conditions, friction model, mesh separation model, description of the criterion, etc. In chapter 3.2. Chapter 3.2 should also present the validation of the FEM model with the experiment. It is not enough to show colored drawings. On what basis was the FEM model verified, e.g. cutting force and its components, material strengthening, chip shape and thickness, length, internal stress, temperature, etc.
- how were the parameters of the Johnson-Cook model determined? Were the conditions from the other experiment identical?
- What were the values ​​of the heat transfer coefficients in the FEM thermo-mechanical model?
- in table 1 please enter the percentage changes in radius and angle. The value one tenth is a small value. However, given as a percentage change is already a large value. This will allow a better representation of the impact of changing specific geometric parameters.
- the caption under figure 1 does not refer to (a), (b), (c).
- chapter 3.3 does not have the correct number.
- figures 2 and 3 can be combined. Figure 3 alone adds nothing new and special.
- own stresses in Figure 6 are presented in newtons! This is a serious mistake. In addition, what components of internal stress were measured? In what place, on what surface. What was the number of repetitions of the experimental measurement.
4. Results and discussion.
- what do the colors show in the drawing? Why is there no scale?
- in Chapter 3, you must first provide accurate model validation.
- what was the size of the chip mesh element. It affects the determination of the chip radius.
- how the radius was determined - it is not enough to give the parameter itself in the drawing!
- in figure 10, please check the correctness of the parameter units
- what stress component do the results in Figure 17 refer to?
5. Conclusions
All the most important results should be summarized in this chapter.
Author Response

(The authors gave the same response as above.)

Round 2
Reviewer 1 Report
The paper can be accepted in the present form
Author Response
Dear Reviewer:
Thank you for your letter and for the reviewers’ comment concerning our manuscript entitled “Influence of Tool Rake Surface Geometry on the Hard Turning Process of AISI52100 Hardened Steel” (ID: materials-580481).
We are very grateful to the reviewers for their recognition of our revised manuscript.
Reviewer 2 Report
The authors provided a revision of the article "Influence of Tool Rake Surface Geometry on the Hard Turning Process of AISI52100 Hardened Steel". All comments are given adequate answers. The authors have done very significant work to improve the article. The article may be accepted for publication in the journal "Materials". I highly recommend this article.
PS: Small remarks at the last stage:
1. In the Nomenclature section, remove the points after m / min and mm / rev
Instead of vc Cutting speed (m / min.) should be vc Cutting speed (m / min)
Instead of f Feed (mm / rev.) should be f Feed (mm / rev)
2. In conclusion 1 (line 379) it is not clear what is being compared and what is it compared to? Explain this clearly.
3. The list of references must be drawn up in accordance with the requirements of MDPI. For example, in the quote 17, 24, 30, 31, 32 you need to register all the co-authors of the article. Please carefully review all articles in the references list.
Author Response
Dear Editors and Reviewers:
Thank you for your letter and for the reviewers’ comments concerning our manuscript entitled “Influence of Tool Rake Surface Geometry on the Hard Turning Process of AISI52100 Hardened Steel” (ID: materials-580481).
First of all, we are very grateful to the reviewers for their recognition of our revised manuscript, as well as some very meaningful comments. In addition, we have studied the comments carefully and have made corrections which we hope meet with approval. Revisions are marked in Blue in the paper. The main corrections in the paper and the responds to the reviewers’ comments are as flows:
